# Where is the 1-million-year-old ice at Dome A?

Liyun Zhao[1,3], John C. Moore[1,3,4], Bo Sun[2], Xueyuan Tang[2], Xiaoran Guo[1]

[1]College of Global Change and Earth System Science, Beijing Normal University, Beijing 100875, China

[2]Polar Research Institute of China, Shanghai 200129, China

[3]Joint Center for Global Change Studies (JCGCS), Beijing 100875, China

[4]Arctic Centre, University of Lapland, P.O. Box 122, 96101 Rovaniemi, Finland

Correspondence to: John C. Moore (john.moore.bnu@gmail.com)

**Abstract**

Ice fabric influences the rheology of ice, and hence the age-depth profile at ice core drilling sites. To investigate the age-depth profile to be expected of the on-going deep ice coring at Kunlun station, Dome A, we use the depth varying anisotropic fabric suggested by the recent polarimetric measurements around Dome A along with prescribed fabrics ranging from isotropic through girdle to single maximum in a three-dimensional, thermo-mechanically coupled full-Stokes model of a $70\times70$ km$^2$ domain around Kunlun station. This model allows to simulate the near basal ice temperature and age, and ice flow around the location of the Chinese deep ice coring site. Ice fabrics and geothermal heat flux strongly affect the vertical advection and basal temperature which in consequence controls the age profile. Constraining modeled age-depth profiles with dated radar isochrones to 2/3 ice depth, the surface vertical velocity, and also the spatial variability of a radar isochrones dated to 153.3 ka BP, limits the age of the deep ice at Kunlun to 649-831 ka, a much smaller range than inferred previously. The simple interpretation of the polarimetric radar fabric data that we use produces best fits with a geothermal heat flux of 55 mWm$^{-2}$. A heat flux of 50 mWm$^{-2}$ is too low to fit the deeper radar layers, and a heat flux of 60 mWm$^{-2}$ leads to unrealistic surface velocities. The modeled basal temperature at Kunlun reaches the pressure melting point with a basal melting rate of 2.2-2.7 mm yr$^{-1}$. Using the spatial distribution of basal temperatures and the best fit fabric suggests that within 400 m of Kunlun station, 1 million-year old ice may be found 200 m above the bed, and there are large regions where even older ice is well above the bedrock within 5-6 km of the Kunlun station.

## 1. Introduction

Finding a continuous and undisturbed million-year old ice core record in Antarctic has been identified by the International Partnership for Ice Core Sciences (IPICS) as one of the most important scientific challenges in ice core research in the near future (http://www.pages.unibe.ch/ipics/white-papers). This is because the last 8 glacial cycles are characterized by irregular cycles of roughly 100 ka in length. Both climate and greenhouse gases co-vary closely. Between 900 ka and 1.2 Ma BP glacial cycles are more regular and are paced at significantly higher frequencies (Lisiecki and Raymo, 2005). The relationship between greenhouse gases and ice sheet growth and decay

during these times is presently unknown since it can only be derived from the atmospheric record archived in an Antarctic ice core covering this time interval.

The search for a continuous and undisturbed stratigraphic record containing 1 million-year old ice has also interested and challenged the ice modeling communities for several decades (e.g., Van Liefferinge and Pattyn, 2013). Potential sites require thick ice, low accumulation rate and cold (that is frozen) basal conditions. However, thick ice increases basal temperatures and may lead to basal melting. Geothermal heat flux is largely unknown in Antarctica and estimates have relatively large uncertainty (Van Liefferinge and Pattyn, 2013), which in turn is the major uncertainty of determining the basal thermal state. Basal temperature is very sensitive to geothermal heat flux, and potentially variable locally in mountainous terrain (e.g. Van Liefferinge and Pattyn, 2013), and localized basal melting and freezing then strongly affects vertical velocity and the age profile (e.g. Sun et al, 2014; Parrenin et al., 2017). Ice fabric is also an important factor in determining the speed of vertical advection in the ice sheet which consequently controls both the basal temperature and the age profile. Depth-varying ice fabric will especially influence the age profile of the deeper ice layers where the base is frozen, although the fabric will not strongly change the temperature profile in the ice. Van Liefferinge and Pattyn (2013) suggested that the most likely oldest ice sites are situated near the divide areas (in some cases, close to existing deep drilling sites, but in areas of smaller ice thickness) and across the Gamburtsev Subglacial Mountains.

Dome A is the top of the East Antarctic ice sheet and above the underlying Gamburtsev Mountains. Being near the center of East Antarctic, at an altitude of about 4092 m a.s.l. the mean annual temperature (that is the measured temperature 10 m below the surface) at Dome A is -58.5°C, the lowest mean annual surface temperature on the Earth (Hou et al., 2007). Ice flow in this region is very slow and less than 0.3 m $yr^{-1}$ (Yang et al., 2014). The average snow accumulation rate is small, about 25 mm ice equivalent $yr^{-1}$ over the past several centuries (AD 1260–2004) (Jiang et al., 2012). Therefore, the Dome A region has good potential for recovery of the oldest ice in an ice core (e.g. Xiao et al., 2008).

Kunlun station (80° 25′ 01"S, 77° 06′ 58"E, 4087 m a.s.l.) was located where the ice thickness is maximal in the vicinity of Dome A specifically for deep ice core drilling to acquire high-resolution records approaching 1 million years in length (Cui et al., 2010). But the mountainous terrain of the Gamburtsev Mountains causes basal melting and refreezing in some places (Bell et al., 2011), which may lead to the loss of the oldest ice, and also complicates the stratigraphic record.

Sun et al. (2014) modeled Dome A ice flow, temperature and age by applying a full-Stokes model to the summit region where detailed surface radar profiles are available, and we use the same domain here. As ice fabric information was not available, Sun et al. (2014) used some simple formulations to define an envelope of possible fabric effects: isotropic and prescribed anisotropic ice fabrics that vary the evolution from

isotropic to single maximum at 1/3 or 2/3 depths. Using these fabrics resulted in basal ages varying by 500 000 years despite age-depth profiles being constrained by dated radar isochrones in the upper one third of the ice sheet. However, Wang et al. (2017) recently presented spatial variations in ice fabric across Dome A obtained from polarimetric radar data in a 30×30 km$^2$ grid around Kunlun Station. Four distinct ice fabric layers were identified and their ages at Kunlun Station found by tracing dated internal ice-sheet layering from the Vostok ice core drilling site.

In this study, we utilize the observed ice fabric determined at Kunlun Station along with several prescribed alternative anisotropic ice fabrics in a three-dimensional, thermo-mechanically coupled full-Stokes model to simulate age-depth profiles, improving the results of Sun et al. (2014). We also use the more plentiful recent measurements including dated radar isochrones at Kunlun station to elucidate the stability of the region on glacial timescales, and the localized variability in geothermal heat flux. Our approach contrasts with that recently used to explore possible ancient ice around Dome C (Parrenin et al., 2017) where a 1D flow model was used in conjunction with extensive radar profiles.

**2. Domain, Data and Mesh**
The surface and topography ice thickness in the Dome A region come from both airborne and ground based measurements. The Antarctic Gamburtsev Province Project (AGAP) surveyed the region with flight lines 5 km apart orientated in the north-south direction (Bell et al., 2011) with perpendicular lines every 33 km. Ground-based surveys were done by the 21st and 24th Chinese National Antarctic Research Expedition (CHINARE) in a 30×30 km$^2$ square along lines typically a few km apart, the along-track radar resolution is 125 m (Sun et al., 2009; Cui et al., 2010). A stake network was also established for ice motion using differential GPS receivers, and data collected in 2008 and 2013 (Yang et al., 2014). More recently polarimetric radar observations were also collected on 5 km-spaced ground-based survey grid (Wang et al., 2017) using a 179 MHz radar using orthogonal orientated antennae with 17 m along-track spacing. They deduced the existence of 4-6 layers of different ice fabric in their survey, and we make use of these data for the ice dynamics simulation.

The Vostok ice core provides absolute dates for radar internal reflections, and since these radar reflections are often continuous over hundreds of kilometers that can provide age-depth profiles over an extensive region of the ice sheet (e.g. Wang et al., 2017). Isochrones in two 150 MHz airborne radar transects collected by the Alfred Wegener Institute were tracked from Vostok to Dome A (Sun et al., 2014; Wang et al., 2017; Fig. 1A) providing 12 dated layers at Kunlun station. We select the 153.3ka isochrones for detailed analysis making use of polarimetric radar data collected in a set of 4 triangles centred on Dome A and 160 km in length.

The modeled domain is a 70×70 km$^2$ square centred at Kunlun station (Fig. 1). The surface and bedrock topographic data in the 70×70 km$^2$ domain come from AGAP while

in the 30×30 km² domain we combined the AGAP and CHINARE data. Crossover analysis of radar lines shows 96% of differences in both surface and bed elevations were less than 150 m. The surface is flat but the bedrock has gradients in excess of 20% (Fig. 1). The domain was divided into 21 vertical layers with the lower 6 having logarithmic spacing with the bottommost layer representing 0.3125 % of ice thickness. The mesh contains 48811 elements and 51940 nodes.

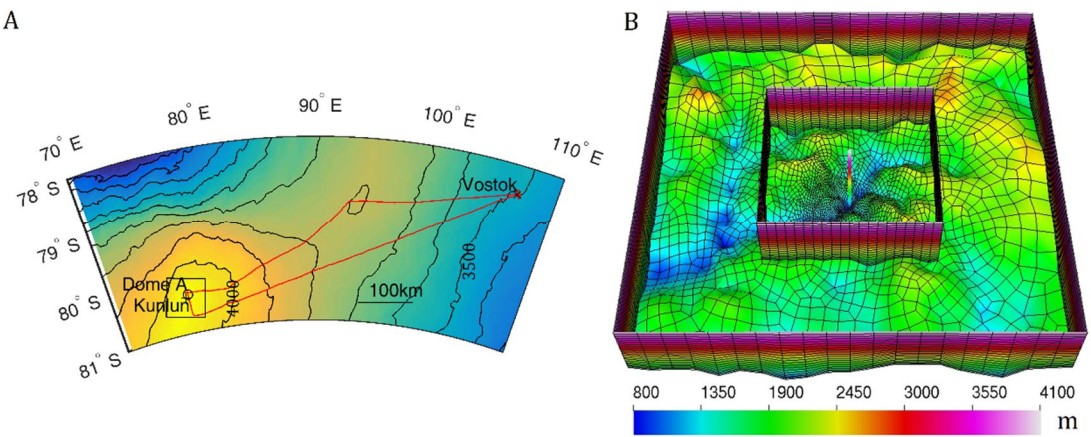

Fig. 1 (A) The locations of Dome A (black circle), Kunlun (black +), Vostok (black ×) and the 70×70 km² study region (black box). The background is surface elevation with 100 m contour interval. The two radar lines that connect the Vostock drill site to Dome A are shown in red. (B) The 70×70 km² finite element mesh in the vicinity of Dome A projected on a polar stereographic map with standard parallel at 71°S and central meridian at 0°E. The background is bedrock elevation. The boundaries of the inner region and the whole region are shown, with the inner 30×30 km² region centred on Kunlun station has 300 m resolution, and the outer region 3 km resolutions. There are 21 terrain-following vertical layers with thinner layers near the base. The bar in the center denotes the drilling site at Kunlun station.

## 3. Model
### 3.1 Field equations
We used the open source finite element method package Elmer/Ice (Gagliardini et al., 2013; http://elmerice.elmerfem.org) to solve the complete three-dimensional, thermo-mechanically coupled "full-Stokes" model across the model domain (Fig. 1). The following equations define the momentum, mass conservation and temperature of the ice:

$$\rho \nabla \cdot \boldsymbol{\sigma} + \rho \boldsymbol{g} = 0, \tag{1}$$

$$\nabla \cdot \boldsymbol{u} = 0, \tag{2}$$

$$\rho c \left( \frac{\partial T}{\partial t} + (\boldsymbol{u} \cdot \nabla)T \right) = \mathrm{div}(\kappa(T)\nabla T), \tag{3}$$

Eqn. (1) is the Stokes equation denoting the balance for linear momentum, the acceleration (inertia force) is negligible, the Cauchy stress tensor $\boldsymbol{\sigma} = \boldsymbol{\tau} - p\mathbf{I}$, where $\boldsymbol{\tau}$ is the deviatoric stress tensor that has a non-linear constitutive

relationship with the strain rate tensor $\dot{\varepsilon} = \frac{1}{2}(\nabla \boldsymbol{u} + \nabla \boldsymbol{u}^{\mathrm{T}})$ that will be discussed

in more detail below, $p$ is the pressure and $\mathbf{I}$ is the identity matrix. Eqn. (2) is

the incompressibility condition which implies the conservation of mass. Eqn. (3) is the heat transfer equation which comes from the conservation of energy.

$\boldsymbol{u}$ and $T$ denote ice flow velocity and ice temperature, $\rho, c$ and $\kappa$ are

density, heat capacity and heat conductivity of ice, $\boldsymbol{g}$ is acceleration due to gravity. We neglect strain heating of the ice by internal deformation.

The age of the ice, $A$, at any point in the ice is governed by

$$\frac{\partial A}{\partial t} + \boldsymbol{u} \cdot \nabla A = 1. \tag{4}$$

### 3.2 Rheology

We use a non-linear anisotropic constitutive relation between the deviatoric stress tensor $\tau$ and strain rate tensor $\dot{\varepsilon}$ following Gillet-Chaulet et al. (2006) and Martin and Gudmundsson (2012),

$$\dot{\varepsilon} = \frac{1}{2\eta_0}\left(\beta\tau + \lambda_1 a^{(4)} : \tau + \lambda_2(\tau \cdot a^{(2)} + a^{(2)} \cdot \tau) + \lambda_3(a^{(2)} : \tau)\mathbf{I}\right), \tag{5}$$

where $a^{(2)}$ and $a^{(4)}$ are the second and fourth-order orientation tensors of ice fabric, respectively, $\mathbf{I}$ is the identity matrix, the symbols $\cdot$ and $:$ are the contracted product and the double contracted product, the three $\lambda$ are expressed as

$$\lambda_1 = 2\left(\beta\frac{\gamma+2}{4\gamma-1}-1\right), \quad \lambda_2 = 1-\beta, \quad \lambda_3 = -\frac{1}{3}(\lambda_1 + 2\lambda_2). \tag{6}$$

The parameter $\beta$ is the ratio of the shear viscosity parallel to the basal plane to that in

the basal plane, and it should be significantly smaller than 1 since ice crystals deform

mainly by shear in the basal plane. The parameter $\gamma$ is the ratio of the viscosity in

compression or traction along the c-axis to that in the basal plane, and it is close to 1

(Gillet-Chaulet et al., 2006). $\eta_0$ denotes the basal shear viscosity,

$$\eta_0 = \frac{1}{2} A(T)^{-\frac{1}{n}} \left(\frac{1}{2}\mathrm{tr}(\dot{\varepsilon}^2)\right)^{\frac{1-n}{2n}}, \tag{7}$$

where $n$ is the power-law exponent and taken as 3, "tr" denotes trace and $A(T)$ is the rate factor described by the Arrhenius law (Cuffey and Paterson, 2010),

$$A(T) = A_0 \exp\left(-\frac{Q}{RT_h}\right), \tag{8}$$

here the coefficient $A_0$ is the prefactor, which takes $3.985 \times 10^{-13}$ Pa$^{-3}$ s$^{-1}$ at temperatures below $-10$ $^\circ$C and $1.916 \times 10^3$ Pa$^{-3}$ s$^{-1}$ at temperatures between $-10$ ℃ and $0$ ℃ ; $T_h$ denotes Kelvin temperature adjusted for melting point depression: $T_h = T + \beta p$ where

$\beta = 9.8 \times 10^{-8} \, \mathrm{KP}_a^{-1}$; Q denotes the activation energy for creep, which takes 60 kJ mol$^{-1}$ at temperatures below $-10$ $^\circ$C, and 139 kJ mol$^{-1}$ at temperatures between $-10$ ℃ and $0$ $^\circ$C; $R = 8.314$ J mol$^{-1}$ K$^{-1}$ is the gas constant.

In Eqn. (5), $\boldsymbol{a}^{(2)}$ and $\boldsymbol{a}^{(4)}$ are defined as

$$\boldsymbol{a}^{(2)} = \oint f(\boldsymbol{c}) \boldsymbol{c} \otimes \boldsymbol{c} d\boldsymbol{c} = <\boldsymbol{c} \otimes \boldsymbol{c}>,$$

$$\boldsymbol{a}^{(4)} = \oint f(\boldsymbol{c}) \boldsymbol{c} \otimes \boldsymbol{c} \otimes \boldsymbol{c} \otimes \boldsymbol{c} d\boldsymbol{c} = <\boldsymbol{c} \otimes \boldsymbol{c} \otimes \boldsymbol{c} \otimes \boldsymbol{c}>, \qquad (9)$$

where $f(\boldsymbol{c})$ is the normalized orientation distribution function (ODF) of the c-axes $\boldsymbol{c}$ with $\oint f(\boldsymbol{c}) d\boldsymbol{c} = 1,$ therefore, the sum of the diagonal components of $\boldsymbol{a}^{(2)}$ equals 1.

In order to reduce the number of variables, we use the invariant-based optimal fitting

closure approximation (IBOF) proposed by Chung and Kwon (2002), the components of $\boldsymbol{a}^{(4)}$ are approximated as functions of those of $\boldsymbol{a}^{(2)}$,

$$
\begin{aligned}
a_{ijkl}^{(4)} = & \beta_1 Sym(\delta_{ij}\delta_{kl}) + \beta_2 Sym(\delta_{ij}a_{kl}^{(2)}) \\
& + \beta_3 Sym(a_{ij}^{(2)}a_{kl}^{(2)}) + \beta_4 Sym(\delta_{ij}a_{km}^{(2)}a_{ml}^{(2)}) \\
& + \beta_5 Sym(a_{ij}^{(2)}a_{km}^{(2)}a_{ml}^{(2)}) + \beta_6 Sym(a_{im}^{(2)}a_{mj}^{(2)}a_{kn}^{(2)}a_{nl}^{(2)}),
\end{aligned} \qquad (10)
$$

where "$Sym$" denotes the symmetrical part of its argument and $\beta_i$ are six functions of the second and third invariants of $\boldsymbol{a}^{(2)}$. Following Chung and Kwon (2002), we

assume $\beta_i$ are polynomials of degree 5 in the second and third invariants of $\boldsymbol{a}^{(2)}$ and use the coefficients computed by Gillet-Chaulet et al. (2006) so that the computed $\boldsymbol{a}^{(4)}$ by (9) fits the fourth-order orientation tensor given by the ODF by Gagliardini and Meyssonnier (1999).

**3.3 Ice fabric**

There are several typical types of fabric in the ice sheet: random ice-crystal fabric, perfect single pole (or single maximum), and vertical girdle fabric. The evolution of the fabric depends on the specific history of stress conditions experienced by the ice as it

travels through the ice sheet. The fabric is represented by the three eigenvalues of the orientation tensor (e.g. Martín and Gudmundsson, 2012), $a_{11}, a_{22}, a_{33}$. Sun et al. (2014) used three simple fabric distributions, but here we include radar observations of fabric

to produce the following 4 archetypes of fabric in the central 30×30 km² domain:

(1) Isotropic fabric (random ice-crystal fabric): $a_{11} = a_{22} = a_{33} = \frac{1}{3}$;

(2) Single maximum (perfect single pole): $a_{11} = a_{22} = 0$, $a_{33} = 1$;

(3) "Girdle fabric" meaning a smooth linear transition from isotropic at the surface to single maximum at some transition depth, $z_s$. Sun et al., (2014) used $z_s$ = 1/3 and 2/3 depth, and thence to the ice base;

(4) "Kunlun fabric" meaning using measured ice fabric layer depths at Kunlun Station. Wang et al. (2017) defined 6 layers for the Kunlun fabric. Here we experimented with subsets of layers.

The Wang et al. (2017) lowermost T5 and T6 layers are rather weak and indistinct in most of the survey grid. The layer T1 is relatively flat, T2 are relatively flat over half the region but has large spatial variation in depth over the other half, while the T3 and T4 layers have large spatial variation of depth and are even missing in some locations around their survey grid. Experiments with 4 layers T1:T4 show a slightly larger vertical velocity at Kunlun station and deeper depths for the 153.3ka isochrone (Section 4.2), and almost the same horizontal flow velocity as with just the top two layers T1 and T2 in our simulations. So here we present simulation results based on a fabric model using just the two upper layers. At Kunlun station T1 is present from the surface to 807.3 m depths, corresponding to ages of 0-57 ka and T2 from 807.3 -1226.2 m with ages of 57-106 ka. The ice is isotropic in T1, then we assume a linear transition from isotropic at the T1 interface with T2 to single maximum at the base of the T2 layer. We then use single maximum for all ice below T2. Wang et al. (2017) do not present unique solutions for the fabric variation in their layers, nor define how the transition from isotropic to single maximum occurs with depth, so the assumptions we make here are perhaps the simplest, but not the only possible interpretations of the fabric data. The three eigenvalues of the orientation tensor for the fabric archetypes we examine are shown in Fig. 2.

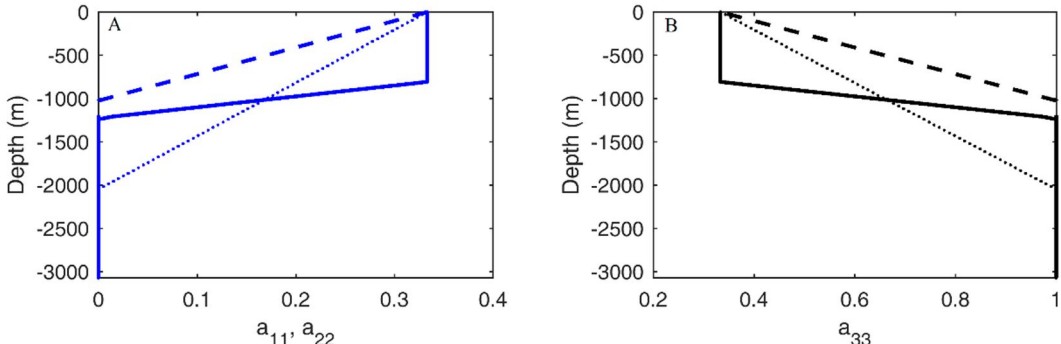

Fig. 2 Fabric as a function of depth (at Kunlun station) for Girdle fabric with $z_s$ = 1/3 (dashed curve), and $z_s$ = 2/3 (dotted curve), and Kunlun fabric (solid curve). The

values of $a_{11}$ (equals $a_{22}$) are in blue in plot (A), while $a_{33}$ is in black in plot (B).

For single maximum $a_{11} = a_{22} = 0,\ a_{33} = 1$; and for isotopic ice $a_{11} = a_{22} = a_{33} = \dfrac{1}{3}$.

### 3.4 Boundary conditions

The ice surface is assumed to be stress-free and changes in atmospheric pressure and wind stress are neglected,

$$\boldsymbol{\sigma} \cdot \boldsymbol{n}\ |_{\mathrm{surface}} = 0 \ , \tag{11}$$

where $\boldsymbol{\sigma}$ is the Cauchy stress tensor and $\boldsymbol{n}$ the unit normal vector pointing outwards.

The present-day surface temperature is −58.5°C, while it is likely about 10°C warmer than that during the Last Glacial Maximum (LGM) over the East Antarctic plateau (Ritz et al., 2001). The viscosity of the ice would change over time and that is not a linear function of temperature. Sun et al. (2014) found that none of the simulations using a surface temperature of -68.5°C matched well with the dated radar isochrones at Kunlun
station, and we confirm that with the extended set of dated isochrones extending to 2/3 ice depth. While glacial period temperatures were likely warmer on average than -68.5°C they were certainly colder than present day. Sun et al. (2014) explain the poor fits for cold surface temperature simulations as being due to key role of warm interglacials in determining the vertical velocity profile of the ice because of the
exponential Arrhenius dependence on temperature of the ice viscosity (Eqn. (8)), along with much higher accumulation rates during interglacials. Therefore we prescribe surface temperature to be the present value of -58.5°C in this study.

We run the model with a no-slip condition at the bed. We could expect that sliding might
occur where there is melting at the bottom. However, surface speeds in our study region is very small (a mean speed of ~11±2.5 cm a$^{-1}$, Yang et al., 2014) and well matched to the model results we show later from ice deformation without basal sliding (Section 4.4) hence basal sliding must be a small fraction of the total velocity, and not affect the results we show.

For a cold base (temperature below the pressure melting point), a Neumann-type boundary condition is applied for the basal temperature,

$$\kappa(T)\nabla T \cdot \boldsymbol{n}\,|_{\mathrm{bed}} = G, \tag{12}$$

where $G$ denotes the geothermal heat flux. For a warm base, (temperature reaching the
pressure melting point), the basal melting rate (i.e. the vertical velocity $W$) is calculated by

$$w = \frac{G - \kappa(T)\nabla T \cdot \boldsymbol{n}\,|_{\mathrm{bed}}}{\rho L}, \tag{13}$$

where $L$ denotes the latent heat of ice.

Geothermal heat flux is the most significant unknown boundary condition. Van Liefferinge and Pattyn (2013) produce a map of the broad-scale heat flux and its uncertainty based on three different estimates, and gives about $50 \pm 25$ mW m$^{-2}$ in the Dome A region. Experiments by Sun et al. (2014) suggest a reasonable spatial pattern of basal melting can be obtained using geothermal heat fluxes in the range of 50-60 mW m$^{-2}$, with values less than about 45 mW m$^{-2}$ producing little or no basal melt in apparent conflict with the radar observations of Bell et al. (2011). Here, we make our simulations with either constant 50, 55 or 60 mW m$^{-2}$ heat fluxes across the domain.

The age of ice at the surface is set to zero. This is not necessarily trivial given the low accumulation rates and low temperatures at Dome A, but there is no evidence from radar that the region was an ablation region (Siegert et al., 2003) with negative accumulation at any time in the past.

At the model domain sidewalls we use an adiabatic boundary (i.e. vanishing normal component) for heat flux and a hydrostatic pressure condition from the surrounding ice.

## 4. Simulations and Results

We did steady-state simulations with present day climate forcing and fixed geometry. We used three values of geothermal heat flux 50, 55 and 60 mW m$^{-2}$, and the 4 different types of fabrics described in section 3.3. The model equations detailed in section 3 were solved numerically with the model Elmer/Ice. In detail, we first computed an isotropic steady-state solution of the velocity and temperature fields for a linear rheology (power-law exponent n = 1 in Eqn. (7)). Secondly, we used these results as initial conditions for an isotropic fabric steady-state run with n = 3. Thirdly, the isotropic results were used as initial conditions for each of the anisotropic fabric steady-state runs. Finally, the age equation was solved and integrated for 1.5 million years by a semi-Lagrangian method (Martín and Gudmundsson, 2012), using the previously obtained steady-state velocity profile.

We first ran simulations with a geothermal heat flux of 50 mW m$^{-2}$, then using a restart from that thermal condition, for the second set of simulations with a geothermal heat flux of 55 mW m$^{-2}$, and then with 60 mW m$^{-2}$.

## 4.1 Modeled age at Kunlun Station

We define a best fit in age profile by the RMS age error of the simulations from the dated radar isochrones. In Fig. 3 we plot these best fit fabrics for each of the 3 geothermal heat fluxes. In addition to the age error we can also usefully estimate model performance by the surface vertical velocity. At steady-state, surface vertical velocity equals surface accumulation rate. The average accumulation during the past 800 ka is 17.7 mm i.e.a$^{-1}$ using the EPICA Dome C record (Bazin et al., 2013), which is very close to what the three best fit simulations achieve (Fig. 3B; Table 1).

With geothermal heat flux of 50 mW m$^{-2}$, the best fit is a girdle fabric with $z_s = 2/3$. The modelled age–depth profile is a noticeably poor fit with the deeper radar isochrones although it matches well in the shallow part. With a geothermal heat flux of 55 mW m$^{-2}$, the simulation using Kunlun fabric is the best fit; and with 60 mW m$^{-2}$, the simulation using a girdle fabric with $z_s = 1/3$ is best. Furthermore, this 60 mW m$^{-2}$ girdle fabric $z_s = 1/3$ is the best match overall to the measured data and gives a basal age of 687 ka (Table 1).

We want to bracket the possible age-depth profile, and make best use of the polarimetric radar observations of fabric. Therefore we use the simulation with Kunlun fabric and geothermal heat flux of 55 mW m$^{-2}$ as an upper bound of basal age (831 ka). For the lower bound we choose the measured Kunlun fabric with geothermal heat flux 60 mW m$^{-2}$ because the lower geothermal heat fluxes seem to produce poor fits while this simulation nicely brackets the best fit overall, although the simulated surface vertical velocity is higher than expected (Table 1). Using this gives a lower bound on basal age of 649 ka.

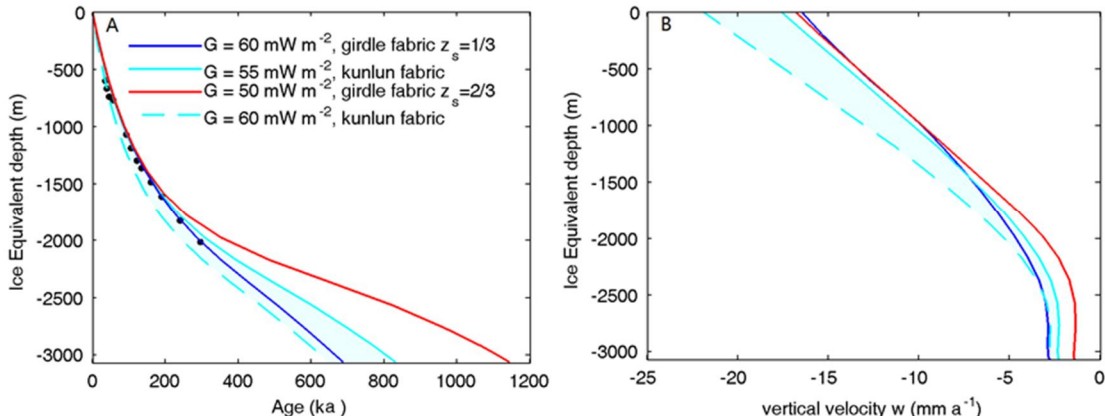

Fig. 3 The best-fit simulations (solid lines) at Kunlun station using geothermal heat fluxes of 50, 55 and 60 mW m$^{-2}$. Modeled age–depth profile (A) and vertical velocity – depth profile (B). The black points denote the dated radar internal reflection horizons tracked from the Vostok ice core site, using a 37 m firn correction (based on the EDML ice core density profile, Ruth et al., 2007) subtracted from the radar depths to convert to the ice-equivalent model scale. The shaded cyan band shows an envelope of acceptable fits to the radar isochrones and age profile with depth, but the dashed (60 mW m$^{-2}$) line likely has too high surface velocities.

Table 1. Modeled and observed isochrones ages and surface velocities at Kunlun

| Simulation | Isochrone age RMS error (ka) | Modeled-observed vertical surface velocity (mm a$^{-1}$) | Horizontal speed RMS error (cm a$^{-1}$) | Modeled age at bedrock (ka) |
|---|---|---|---|---|
| Kunlun fabric, G=55 mW m$^{-2}$ | 10.85 | -0.1 | 4.9 | 831 |
| Kunlun fabric, G=60 mW m$^{-2}$ | 26.00 | 4.2 | 2.7 | 649 |
| Girdle fabric $z_s = 1/3$, | 6.88 | -1.2 | 3.9 | 687 |

| | | | | |
|---|---|---|---|---|
| G=60 mW m$^{-2}$ | | | | |
| Girdle fabric $z_s = 2/3$, G=50 mW m$^{-2}$ | 27.18 | -0.9 | 6.9 | 1143 |

## 4.2 Spatial variability of fabric

We examine how the spatial variation in depth of the 153.3 ka radar isochrone along a track centered at Dome A and passing Kunlun station (Fig. 4A) can be simulated with
the fixed fabrics that define the best fits in Fig. 3. We define misfit using a robust measure that is by the median of the absolute difference between the modeled and measured depths. Fig. 4B shows that among the three best fit simulations, the 50 mW m$^{-2}$ simulation has the largest misfit of 360 m, while the misfit of the other simulations are all less than 180 m, with the best overall fit (93 m) using the lower bound basal age
simulation of Kunlun fabric with $G$=60 mW m$^{-2}$.

Wang et al. (2017) observed large spatial variability of the depth of the ice fabric layers T2:T4. As noted in Section 3.3, we applied the depth of the top two fabric layers T1:T2 at Kunlun station to the whole region. Wang et al. (2017) show that the depth of the T2
layer is relatively constant in the region to the north of Kunlun Station (which includes triangle 4 in Fig. 4A), but is much deeper in the southern region (including triangles 2, and 3). Smaller T2 layer depth would result in slower simulated ice vertical velocity, hence older ice. Therefore, the modeled depths of the 153.3ka isochrones to the south of Kunlun are underestimated. The underestimation is larger where the ice is thicker
(triangle 3) and smaller in triangle 2 where the ice is thinner, see Fig. 4B.

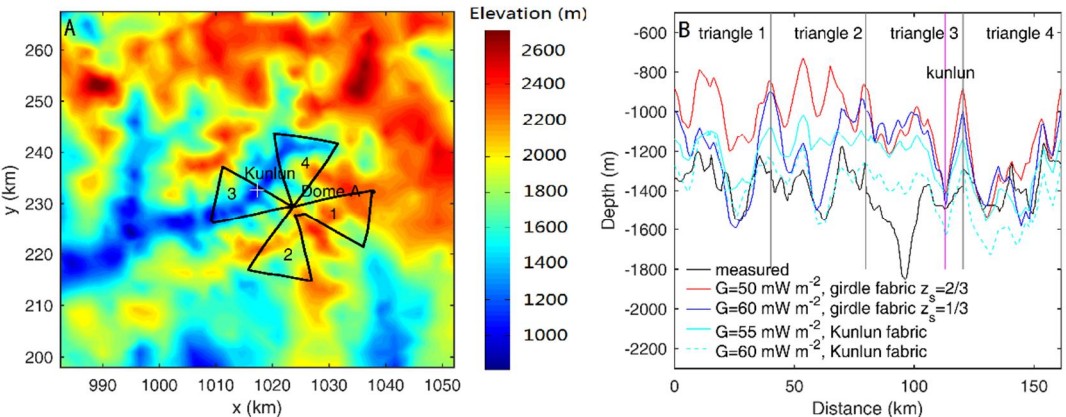

Fig. 4 (A) Measurement tracks (black curve) consisting of 4 triangles, for the depth of the 153.3 ka isochrone layer; the common point of the four triangles is Dome A; the white cross is Kunlun station, the background is bedrock elevation in $70 \times 70$ km$^2$
region; (B) the measured (black) and modeled (colored) depths of the 153.3 ka isochrone using the simulations shown in Fig. 3. The distance coordinate in (B) starts from Dome A and follows the tracks of triangles 1-4, with Dome A passes marked by vertical black lines, and Kunlun Station is marked as a vertical magenta line in triangle 3.

## 4.3 Modeled age at depth in the central region

Using measured Kunlun fabric and geothermal heat fluxes of 55 and 60 mW m$^{-2}$, the modelled age at 95% depth in the central $30 \times 30$ km$^2$ region is shown in Fig. 5. The age dependence on ice depth is such that deep ice that melts has relatively young ages at 95% depth, and so also does thin ice. Melting removes old ice at the base, while thin regions have all their very old ice very close to the bed. There are many more locations where the age simulation reaches the 1.5 Ma limit under the 55 (Fig. 5C) than under a 60 mW m$^{-2}$ (Fig. 5D) heat flux reflecting the more widespread basal melting. The maximum age is reached at depths as shallow as 2000 m under both heat fluxes (Fig. 5B), showing that a shrewd (or lucky) choice of location may recover very ancient ice even under the higher heat flux. But there are no locations with the oldest ice at depths above 2600 m with the 60 mW m$^{-2}$ heat flux, and above about 2800 m with 55 mW m$^{-2}$ heat flux (Fig. 5B). As we discussed earlier in section 4.2, the age in triangle 3 are probably overestimated (Fig. 5C and D). The modeled age inside triangle 4 has the most confidence.

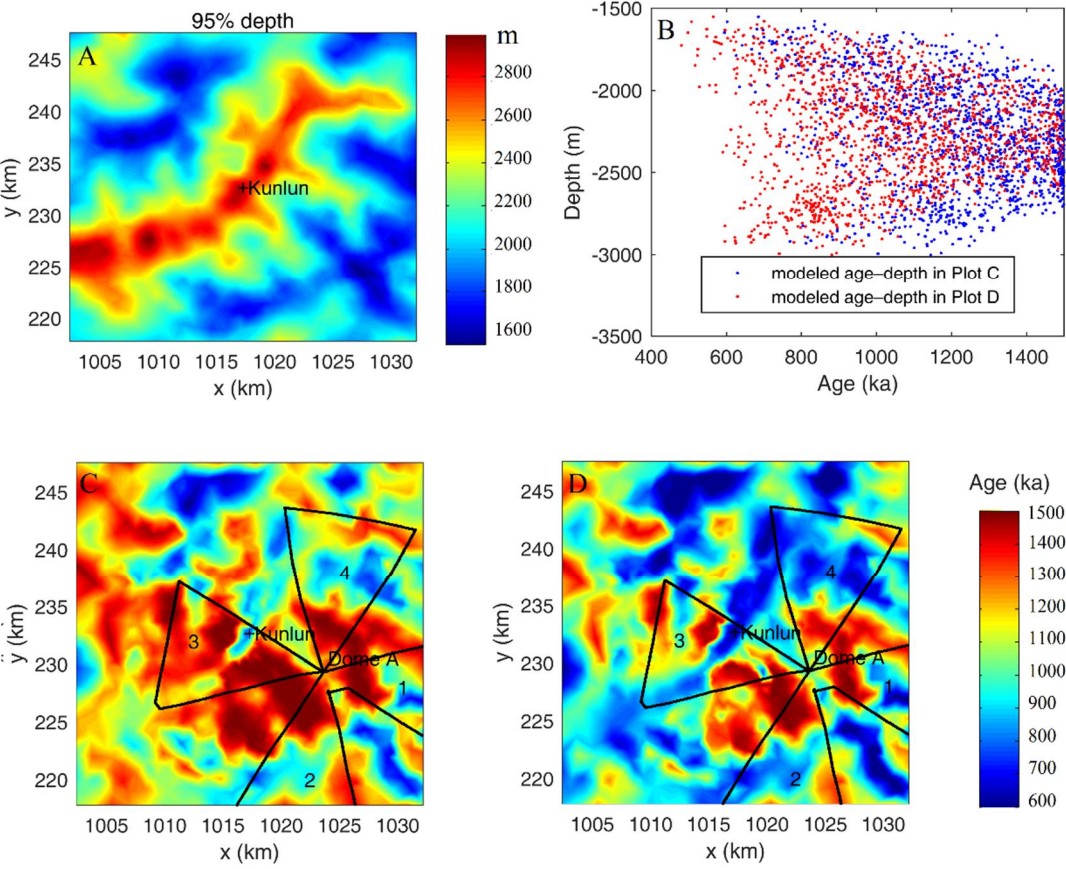

Fig. 5. 95 % depth in the central $30 \times 30$ km$^2$ model domain (A) and modeled age of the ice at this depth using Kunlun fabric and a geothermal heat flux of 55 (C, blue dots in plot B) and 60 (D, red dots in plot B) mW m$^{-2}$ and surface temperature of −58.5°C. The areas with no basal melt are arbitrarily limited to an age of 1.5 Ma. The black cross is Kunlun station. The black line marks the polarimetric radar survey route triangles marked in Fig. 4.

At the Greenland summit drill site, the GRIP ice core contains small (cm-scale) overturned folds 200 m above bedrock (Taylor et al., 1993), at Dome C stratigraphic continuity was lost only 60 m above the bed (Tison et al., 2015). Although the bedrock topography is smoother in central Greenland than around Dome A, ice sheet temperatures are warmer, vertical velocities higher and the potential of summit

migration over glacial cycles probably greater than the Dome A region. The GRIP ice core is in a similar dynamical pure stress (vertical compression-only) regime as Dome A, but it is not a perfect analogy. Dome C may be a better analogy but as a conservative approach we map the age of the ice 200 m above bedrock in Fig. 6. There is ice at least 1 million years old ice simulated on the side slopes of the valley below Kunlun station.

The closest to Kunlun station being found directly below a point about 380 m away under 55 mW m$^{-2}$, and 1 km away under 60 mW m$^{-2}$ heat fluxes. However this position is a less reliable part of the domain than the area to the north of Kunlun in triangle 4, where old ice is about 5 km away.

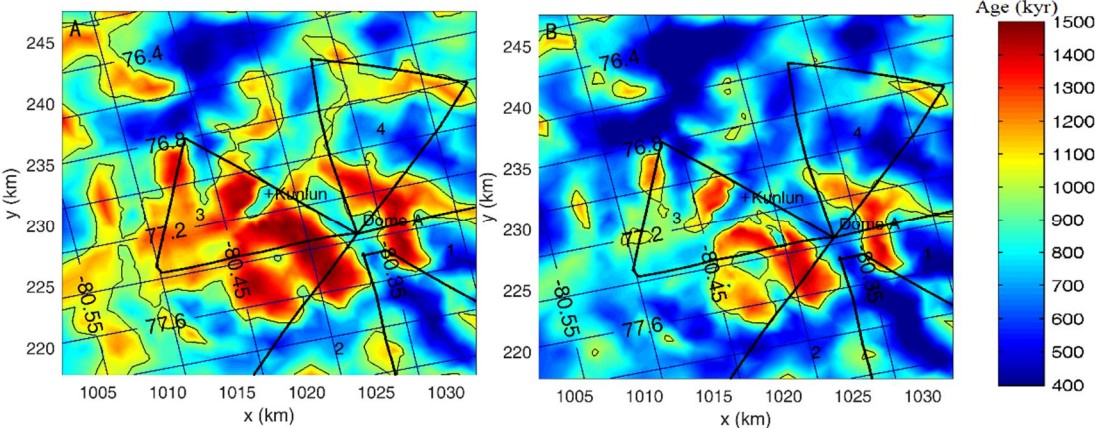

Fig. 6. Modeled age at the height of 200 m above the bedrock using Kunlun fabric and a geothermal heat flux of 55 (A) and 60 (B) mW m$^{-2}$ in the standard grid coordinate system (unit: km, see Fig. 1), and WGS 1984 latitude and longitudes (inclined grid with the South Pole to the lower left) . Kunlun station is marked by a black plus sign. The black curve is the 1 Ma age contour. The thick black line marks the polarimetric radar

survey route triangles marked in Fig. 4.

### 4.4 Modeled surface velocity comparison with observation

Yang et al. (2014) calculated the surface horizontal velocity field at 12 survey stakes around Dome A using repeated GPS measurements, and found a mean speed of ~11±2.5

445 cm a$^{-1}$, with the maximum velocity of 29±1 cm a$^{-1}$ and the minimum surface velocity of 3.1±2.6 cm a$^{-1}$. The modeled surface horizontal velocities from the four best-fit simulations are very similar to each other (Fig. 7A), and are very close to the observed in both magnitudes and directions. There is less variability between the 4 different simulated velocities than with the observed velocities. Thus the fabric cannot be

usefully determined by the horizontal surface velocity components.

The modeled surface vertical velocity distribution is shown in Fig. 7B and, as discussed earlier (Section 4.1), may be compared with local accumulation. Within the central

30×30 km² domain almost all surface velocities are within ±50% of the value at Kunlun station. There are some larger differences near the border of the larger 70×70 km² domain, with small parts even having upward velocities. This is likely an indication of the model transition zone flow to the surrounding ice sheet rather than a real effect.

Local accumulation is associated with precipitation, small scale surface topography over the flat interior of the ice sheet, and wind-driven post-depositional processes (e.g., Frezzotti et al. 2005; Ding et al., 2011). Recent and palaeo-surface accumulation rates across Dome A have been measured (e.g. Hou et al., 2007; Ding et al., 2011) and show that Dome A area has the lowest accumulation rate and smallest spatial variability along a transect from the coast to the summit. This is because it is the coldest and highest region, with smooth topography, furthest from the coast, and has the lowest surface wind speeds. The variations in vertical velocity from the model are not prescribed by surface weather, but determined by mass conservation, and hence reflect advection processes in the ice sheet. Any differences from measured accumulation indicates that the ice sheet is out of steady state balance. As shown in Fig. 3 there are only small differences in vertical velocity for the best fit fabrics for each of the three geothermal heat fluxes we use, though the lower age bound using a 60 mW m$^{-2}$ heat flux produces a too large value at Kunlun. Hence, although the vertical velocity does not in practice constrain the ice fabric, it can help eliminate too high a geothermal heat flux.

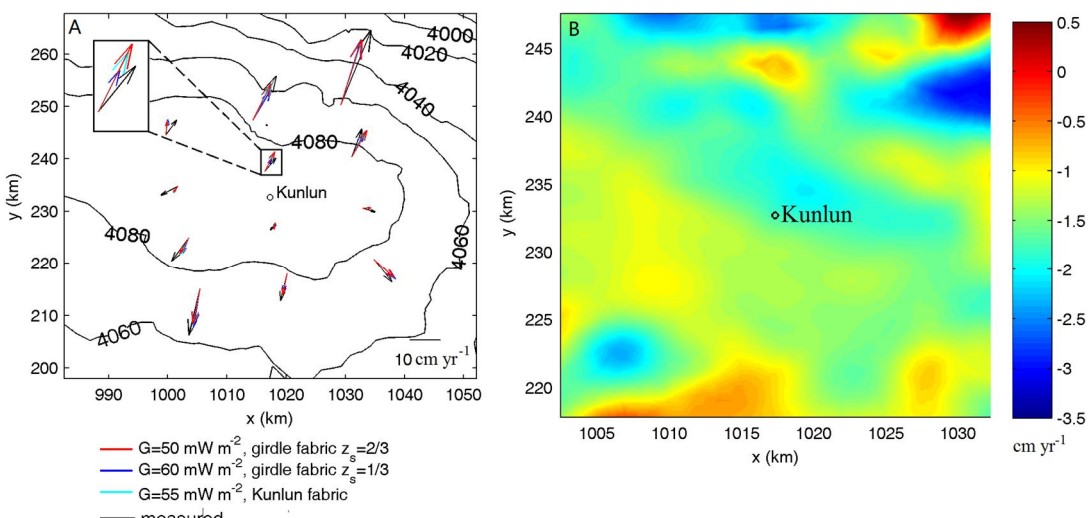

Fig. 7 (A) Surface topography with contours, and the measured (black arrows; Yang et al., 2014) and modeled surface horizontal velocity (see legend for details) near the Kunlun Station. Kunlun station is marked by a black circle. The coordinate system is WGS 1984 plotted using Antarctic Polar Stereographic with standard parallel at 71° S and central meridian at 0°E. The inset box is a zoom-in on one velocity datum showing the differences between ice fabrics. (B) Modeled surface vertical velocities (unit: cm a$^{-1}$) using Kunlun fabric and a geothermal heat flux of 55 mW m$^{-2}$ and surface temperature of −58.5°C. Note the region plotted in panel (B) is the central 30×30 km² area while in (A) it is the larger 70×70 km² region.

## 4.5 Modeled basal melt and temperature

Basal temperature depends on surface accumulation rate, ice thickness and basal geothermal heat flux. Since we use fixed geometry, the surface accumulation rate equals surface vertical velocity. As shown in Fig. 7B, the spatial variation of surface vertical velocity is very small in the central $30 \times 30$ km$^2$ region. Therefore, the high temperature area is located along the valley where the ice is thick (Fig. 8). Using Kunlun fabric and a geothermal heat flux of 55 mW m$^{-2}$, the basal ice at Kunlun station drill site is predicted to be at pressure melting point (Fig. 8A), along with most of the large valley. But there is simulated to be cold basal ice within a kilometer from Kunlun station (Fig. 8A). The spatial extent of melting is considerably larger using geothermal heat flux of 60 mW m$^{-2}$ (Fig. 8B), with several of the side valleys now simulated to melt.

Bell et al. (2011) show extensive melt and refreezing features in the Gamburtsev Mountains. Refreezing is driven by ice thickness gradients pushing water up slope to cooler regions where is can refreeze. This is most likely where a bedrock ridge occurs across the general direction of water flow driven by hydraulic potential. No refreezing features were observed within the domain we model here. Surface slopes in the summit region of Dome A are very low (Fig. 7A), so the hydraulic potential (Fig. 8C) of water at the bed is essentially governed by the bed slopes. Calculation of hydraulic potential shows that is indeed the case and water flow should be along the valley in the vicinity of Kunlun drill site. The oldest ice closest to Kunlun (Fig. 6) is expected perpendicular to this flow direction, on the valley walls or the regions without basal melt in Fig. 8.

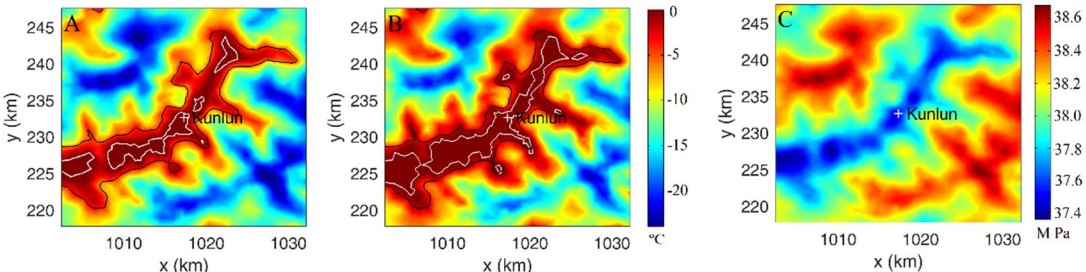

Fig. 8 Basal temperature relative to pressure melting point using Kunlun fabric and a geothermal heat flux of 55 (A) and 60 (B) mW m$^{-2}$, and the hydraulic potential (C). The bedrock areas at pressure-melting point in A and B are surrounded by a white contour. Kunlun station is marked as white plus sign. The black curve in A and B shows the outline of the valley defined as the 1500 m contour in bedrock elevation.

## 5. Uncertainties

Our approach here is relatively sophisticated in terms of ice models presently in use, but there are several limitations that almost certainly mean that details of the simulation will be wrong. We make the key assumption that the ice sheet is in steady-state, and the surface geometry is fixed, which means the surface accumulate rates balances the vertical velocity and it is also fixed in time. However, the basal thermal condition is

sensitive to the ice thickness although other simulations of the whole Antarctic ice sheet suggest that elevation changes at Dome A have been less than 50 m over glacial cycles (Ritz et al., 2001; Saito and Abe-Ouchi, 2010.) Transient simulations with varying geometry and surface accumulation rate in the past 800 ka would improve the model result.

We used a spatially constant geothermal heat flux. Although geothermal flux may over kilometer scales, it seems unlikely in East Antarctica. For example, Carson et al. (2014) suggest heat flow may vary by a factor of >150% over 10–100 km length scales in East Antarctica. Passalacqua et al. (2017) explored variation in heat flux around Dome C using data from radar surveys, and prescribe uniform geothermal heat flux over 10 km scales. Schroeder et al. (2014) similarly infer geothermal heat flux variability from radar surveys over Thwaites glacier in West Antarctica, which is proximal to the Mount Takahe volcano that was active during the Quaternary, finding heat fluxes could double over ranges of about 20 km. We do not expect any recent magmatic activity in the Gamburtsev Mountains, and the situation of Dome C is probably a reasonable analogue. However there is simply no data to constrain heat flux around Dome A, and hence modelled thermal structure, ice viscosity and age-depth profile. Liefferinge and Pattyn (2013) explored the uncertainty in existing geothermal heat flux data sets and their effect on basal temperature with a spatial resolution of 5 km. The basal temperature was calculated using the steady-state thermodynamic equation in which ice flow velocity is calculated from the shallow-ice approximation. The mean geothermal heat flux of the three existing datasets at Dome A is about 45 mWm$^{-2}$, with root mean square error of about 20 mWm$^{-2}$. Their modelled basal temperature at Dome A is about -10°C corrected for the dependence on pressure with a root mean square error of about 6°C. Due to the coarse resolution (5 km) used in the whole Antarctic simulations of Liefferinge and Pattyn (2013), the modelled basal temperature does not have obvious spatial variation across the Dome A region at scales of hundreds kilometers.

The Gamburtsev Mountain is characterized by large spatial variability in bedrock topography, which means that a full-Stokes model that considers the all the stress components is better able to capture the ice dynamics than does the shallow-ice approximation (e.g., Zhao et al., 2013). In our study, large variations in basal temperature are simulated using a full-stokes model run at around 500 m resolution. The basal thermal state is then very sensitive to geothermal heat flux (Sun et al., 2014), which we explored using 45, 50, 55 and 60 mWm$^{-2}$, and which spans the broad range suggested by Liefferinge and Pattyn (2013).

We also use a spatially constant fabric across all our model domain, with transitions between fabrics at two fixed depths taken from those measured at Kunlun station by Wang et al. (2017). As discussed in Section 4.2, this leads to lower confidence in the age of the basal ice in the region south of Kunlun than to the north. This further means that we have more confidence in finding very old ice in the slightly further away northern region of Fig. 6 than to the south of Kunlun.

Our results suggest spatial variability in basal melting, and this may introduce basal accretion in places (Bell et al., 2011), though there is no radar evidence of any basal accretion features in the vicinity, the model could be improved by adding basal hydrology. Basal melting may also introduce sliding at the ice/bed interface, which we explicitly excluded in the model, however, comparison with observed horizontal velocities suggests that this is not an issue. Indeed extraction of sliding rates from inverse modeling using observed velocities would be extremely difficult at Dome A given the very low speeds making satellite interferometry impossible, and the sparse network of GPS locations.

Surface measurements of horizontal velocity do not constrain fabric information in the ice sheet. The influence of fabric is felt in the deeper ice not near the surface. Hence accurate estimates of fabric must rely on observations from the deeper layers, such as radar isochrones, or potentially vertical velocity profiles from phase sensitive radar. These observations together with a flow model allow geothermal heat flux and thence basal temperatures to be estimated over extended regions where assumptions of unchanging heat flux and fabric hold. Testing this hypothesis by tracking the depths of a 150 ka isochrone with the model suggests that fabric and heat flux variations are not very fast on 10 km horizontal scales, but that localized basal melt may complicate this diagnostic method.

The special ice flow conditions at ice divides often leads to the presence of Raymond arches (Raymond, 1983), where older ice is at shallower depth than it is several ice thicknesses away from the divide. These features are visible as uplifted radar internal reflections in profiles across the divide. The strongest Raymond arches show up in high-accumulation coastal domes where the bed is cold and flat and the ice column is closer to isothermal (e.g. Hindmarsh et al., 2011). However, bed topography is complex at Dome A and Raymond arches are not seen in the observed radar profiles. Furthermore, our ice dynamics package, Elmer-ice, includes all the physics needed to produce the Raymond effect, but we also detect no such feature in transects across the flow divide. We explain this by the Raymond arch being obscured by a combination of rugged basal topography and thermal structure. The strong thermal gradient in the ice sheet tends to reduce the Raymond effect: the tendency of the non-Newtonian rheology to produce a stiff layer near the bed where strain rates are low is counteracted by the tendency of warm temperatures to produce softer ice at depth. The viscosity of the basal ice under the dome is softer than the viscosity of the super cold ice near the surface, but it is still much stiffer than the basal ice away from the dome, causing the old ice to be up-warped somewhat under the ridge. Moreover, the high basal melt rates of 2-3 mm a$^{-1}$ at Kunlun station draws down ice and obscure the Raymond effect.

Very old and deep ice near bedrock is likely to have experienced vertical mixing via various mechanisms: boudinage between layers with different rheology, small scale non-laminar flow, or regelation around any bed irregularities (Taylor et al., 1993).

Although in central Greenland mixing was limited to areas closer than 200 m above the bed, mixing may scale with the vertical relief in the area, which would be very large in the case of the Kunlun site if the ice dome location has migrated by 10 km or more over history. However, the coherence of the radar isochrones to at least 2/3 ice depth from Vostok through Gamburtsev mountains to Dome A suggests that vertical mixing to the

topographic scale of the mountains has not occurred. Furthermore analysis of the EPICA Dome C ice core revealed continuous stratigraphy to within 60 m of bedrock (Tison et al., 2015), and Parrenin et al. (2017) use that as a basis for locating ice up to 1.5 Ma old in the Dome C region. Comparing our Fig. 6 with the analysis in Parrenin et al. (2017) shows far more locations having ice at least 1.5 Ma further than 200 m

from the bed in the vicinity of Dome A than at Dome C. The nearest such ice to the Concordia station is about 10 km away, compared with 0.5-5 km from Kunlun station.

## 6. Summary and Conclusions

Using the constraints of observed ice fabric from polarimetric radar observations, depths of dated internal isochrones, along with reasonable estimates of surface vertical velocity allows us eliminate both geothermal heat fluxes lower than 50 and higher than 60 mW m$^{-2}$ at Kunlun station. The lower heat flux together with observed fabric produces poor fits to dated radar isochrones deeper than half ice depth. The higher heat

flux produces too fast a vertical velocity at the surface that is inconsistent with good fits to measured accumulation rate and to the dated isochrones.

       The best fits to the isochrones and surface velocities constrain rather closely the range of basal ages at the Kunlun drilling site to about 650-830 ka, with the upper end more

likely than the lower because the lower age bound comes from an unrealistic 60 mW m$^{-2}$ heat flux. The spatial variability of age at 95% ice thickness illustrates the non-linear dependence on ice thickness. Ice that is too deep lacks old ice due to melting, ice too thin leads to old ice being too close to the bed to be useful for ice coring.

Reasonable ice core stratigraphy may be preserved to 200 m above bed, as is the case in central Greenland, or 60 m in the case of Dome C, so we determined locations having ice at least 1 million years old ice at least 200 m above the bed. Using our favored values for geothermal heat flux and ice fabric, ice this ancient may be found by vertical drilling within 400 m of the present Kunlun drill site, indeed this location would contain

much older ice since it seems to be frozen to the bed. However we have more confidence in our simulation of ancient ice about 5-6 km to the north of Kunlun station than the closer sites to the south. Near-basal ice this close to Kunlun may be accessible with a straight forward repositioning of the drilling site (Talalay et al., 2017) rather than the logistics base. Hydraulic potential suggests that the regions of old ice near Kunlun

would not contain refrozen melt water from the deeper valleys. Multiple cores from the same borehole may be recovered, sampling different climate periods in detail as basal melting effectively stretches the relative younger ice. Thus the Kunlun station is well suited to provide the longest continuous stratigraphic record from Antarctica.

**Acknowledgements**

This study is supported by National Natural Science Foundation of China (Nos. 41506212, 41530748, 41376192) and National Key Science Program for Global Change Research (2015CB953601). We thank T. Zwinger and C. Martín for their advice and the anisotropic fabric and age-depth solver that used in the Elmer/ICE model, M. Wolovick for general insights into the Dome A glaciological setting, and two anonymous referees for their helpful comments.

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
