# Peer review of "Where is the 1-million-year-old ice at Dome A?"

_The Cryosphere, 2017_

## Referee Comment (RC1) · Anonymous Referee #1 · 1 Feb 2018

In this manuscript, the authors model the age field in a 70x70 km region centered around Kunlun Station (Dome A). The modeling is based on the finite-elements code Elmer/Ice. It takes into account the anisotropy of the ice material due to its fabric, the mechanical behavior of the ice and the temperature field. An important assumption is that the ice sheet is in steady-state, so only a steady-state velocity field is computed. Various hypotheses are tested regarding the geothermal flux and the fabric. The model is compared to age observations at Kunlun station obtained by tracing radar layers to the dated Vostok ice core. It is found that the best agreement is obtained with a geothermal flux of 60 mW/m2 and a fabric evolving from isotropic at surface to a girdle fabric at depth=2/3 of ice thickness. From there, the model extrapolates the basal age to the range 650-830 kyr BP at the base of the ice sheet at Kunlun station,

that is too young to record the Mid-Pleistocene Transition (MPT). The model is also compared to horizontal surface velocity measurements, but it is found that it is difficult to discriminate between the different geothermal and fabric assumptions. The surface vertical velocity model is also compared to surface accumulation measurements, which allows to eliminate values of geothermal flux higher than 60 mW/m2. Finally, some locations for old ice recording the MPT are proposed, 1-2 km maximum far from the Kunlun station, on the flanks of a bedrock valley.

Generally, I enjoyed reading this manuscript which is clearly written. The modeling experiments presented are an advance with respect to the state-of-art ice flow and age modeling around Dome A, despite some rough assumptions. However, I have some major concerns explained below:

- there is no discussion on the Raymond effect, which occurs at domes with a non-linear rheology and which has an important influence on the age-depth profile. The Raymond arches should be present in their modeling experiments. In reality, the Raymond arches are probably not easily observable in the radar age observations, since the dome has probably moved during the past (a movement of only a few kilometers is sufficient to dilute the Raymond effect spatially). This is a clear limitation of the steady-state assumption when modeling the age of the ice in the vicinity of a dome. A discussion on this effect is mandatory.

- Why is the age model compared to the radar age observations only at Kunlun? The comparison could be done anywhere where there are radar data.

- if I understood correctly, to compare the modeled vertical velocity with the accumulation observations, the authors use an average accumulation which is calculated as a weighted average of glacial and interglacial accumulations. This is too rough an hypothesis. The authors should use the EPICA Dome C record to calculate a ratio between the present-day accu and the 800 kyr average accu. This way, the comparison with the modeled vertical velocity would be more relevant.

- in a similar way, the authors should use a 800 kyr average value of the surface temperature based on the Dome C temperature variations (assuming the variations are the same at Dome A), rather than simply the present-day value at Dome A.

- there is a mistake at the beginning of section 4.1. At steady-state, surface vertical velocity should be equal to surface accumulation rate, not surface accumulation rate plus basal melting. This should be corrected.

- Because of these rough assumptions in the modeling, a perspective paragraph listing what could be improved in a future modeling study would be welcome.

I also have some minor points below:

- l.51: "Hou et al., 2007" -> missing space

- l.102: "special" -> "spatial"

- fig.1A is difficult to ready. I would use a square region in a classical projection.

- l.165: "is gas constant" -> "is the gas constant"

- l. 173: "the components of..." -> missing space

- l. 242: why not using an intermediate value of the surface temperature between the present-day and the LGM? (Cf. comment above).

- l245: the no sliding assumption is quite rough. There is probably sliding where there is melting.

- l. 261: quite a big assumption here also, since the geothermal flux might change at a kilometer scale.

- l. 326: the reference is Ruth et al., not Urs et al. (Urs is the first name).

---

## Referee Comment (RC2) · Anonymous Referee #2 · 20 Feb 2018

Review of Where is the 1-million-year-old ice at Dome A? Liyun Zhao, John C. Moore, Bo Sun, Xueyuan Tang and Xiaoran Guo

The study aims to locate areas around Kunlun Station that contain ice older than 1Myr, and the manuscript describes results from simulations with the Elmer/Ice model, a 3D, thermomechanical, Full-Stokes, finite-element model. The model parameters are constrained by observations of surface velocity, ice fabric and radar isochrones. The authors test different ice fabrics and geothermal heat fluxes and conclude through the comparison with observations that the geothermal heat flux is likely between 55mW/m2 and 60mW/m2. From the model results, areas that potentially contain ice older than 1Myr are identified based on these two geothermal heat flux values. Overall, the manuscript is well written and is definitely of interest to the glaciological community

especially those concerned with ice-core sites and the retrieval of "the oldest ice". I have some main points that I would like to see addressed, as well as some minor points that might improve readability.

Main points

1 I am missing a more thorough discussion of the uncertainties or rather the confidence the authors have in their results (Figs. 6 and 8). Firstly, a comparison with the results from van Liefferinge and Pattyn (2013) would be interesting in order to give insights into how coarse vs. fine grid, and shallow-ice vs. full-Stokes might affect results. Secondly, the vertical distribution of ice fabric is not in line with the findings of Wang et al. The authors state that they obtain the same results for two or four layers of fabric but the same what? Age-depth distribution? Surface velocity? 153.3kyr isochrone depth? Based on the figures in Wang et al., I would say that an assumption of two layers is fine around Kunlun station but does not appear to be valid further away from the station. Could the variability of T3 and T4 explain the variation in fit with the isochrone? Finally, the fact that the model has a tendency to underestimate the depth of the 153.3kyr isochrones indicates that the model could be overestimating the amount of 1Myr old ice (with the possible exception of triangle 4). Especially triangle 3 is problematic and perhaps by extension the age of the valley.

2 Most of the figures are small and hard to interpret. Please add legend and/or title so that main points of the figures are understandable without consulting the figure caption. The symbol for Kunlun Station, the letters distinguishing the figures and the axes labels are hard to read.

3 The structure of the paper could be improved. For example, the introduction and the discussion/conclusion appear fragmented rather than cohesive sections (e.g., the introduction contains a section about Kunlun, then a section about ice fabric and vertical velocities and then another section about Kunlun). I am also missing an introduction to the datasets in Section 2. The surface velocity data, the polarimetric data and the

isochrones depths should be mentioned in Section 2 before they are used in Section 4.

Minor points

Lines 13-33: The abstract should also state the aim of the study.

Line 13 (and other places): age/depth or age-depth?

Line 25 and 313: Typo "polarimetric"

Lines 35-42: This part of the introduction needs a few more sentences about the background for the search a 1Myr ice, for example, with references to the IPICS white paper and to Fisher et al., 2013 (https://www.clim-past.net/9/2489/2013/)

Lines 56-62: There is no information about the Dome A ice core. Since an ice core has been drilled, is there no information about ice fabric, age-depth profile or temperature from that? Line 69: Is Parrenin et al., 2017, the correct reference here?

Lines 96-107: What is the original resolution of the AGAP data and the Chinese data? How does the resolution compare to that of the model?

Line 130: Word missing?

Line 131: Add that the strain rate tensor will be discussed in more detail below

Line 145: Typo: "Gudmundsson"

Line 157: what is n equal to?

Lines 160-165: These values might be better summarised in a table

Section 4: I would like a table in this section where different simulations are listed with least-square age error, vertical surface velocity, age at the bedrock and the magnitude of the difference between measured and modelled horizontal velocity. This would allow readers to judge the difference between the model results and how well they agree with observations.

Lines 360-364: explicitly refer to the figure that is being discussed (5C, 5D etc.) otherwise it is hard to follow.

Line 402: I agree with reviewer #1 re. the comparison with accumulation + basal melt rate.

Line 442: please indicate on a figure where this valley is.

Line 455: The hydraulic potential should be shown on a figure or in the supplementary material. The description here does not help the reader.

Line 522: Considering that Dome A has very low accumulation rates, how can an ice-core from the area provide the highest resolution record?

Fig. 1: Both figures appear pixelated and especially 1A is not very informative.

Fig. 2: this figure is hard to read, why not split it into two figures?

---

## Author Comment (AC1) · 3 Apr 2018

In the reply, the referee's comments are in *italics*, our response is in normal text, and quotes from the manuscript are in blue.

*In this manuscript, the authors model the age field in a 70x70 km region centered around Kunlun Station (Dome A). The modeling is based on the finite-elements code Elmer/Ice. It takes into account the anisotropy of the ice material due to its fabric, the mechanical behavior of the ice and the temperature field. An important assumption is that the ice sheet is in steady-state, so only a steady-state velocity field is computed. Various hypotheses are tested regarding the geothermal flux and the fabric. The model is compared to age observations at Kunlun station obtained by tracing radar layers to the dated Vostok ice core. It is found that the best agreement is obtained with a geothermal flux of 60 mW/m$^2$ and a fabric evolving from isotropic at surface to a girdle fabric at depth=2/3 of ice thickness. From there, the model extrapolates the basal age to the range 650-830 kyr BP at the base of the ice sheet at Kunlun station that is too young to record the Mid-Pleistocene Transition (MPT). The model is also compared to horizontal surface velocity measurements, but it is found that it is difficult to discriminate between the different geothermal and fabric assumptions. The surface vertical velocity model is also compared to surface accumulation measurements, which allows to eliminate values of geothermal flux higher than 60 mW/m$^2$. Finally, some locations for old ice recording the MPT are proposed, 1-2 km maximum far from the Kunlun station, on the flanks of a bedrock valley.*

*Generally, I enjoyed reading this manuscript which is clearly written. The modelling experiments presented are an advance with respect to the state-of-art ice flow and age modeling around Dome A, despite some rough assumptions. However, I have some major concerns explained below:*

*- there is no discussion on the Raymond effect, which occurs at domes with a non-linear rheology and which has an important influence on the age-depth profile. The Raymond arches should be present in their modeling experiments. In reality, the Raymond arches are probably not easily observable in the radar age observations, since the dome has probably moved during the past (a movement of only a few kilometers is sufficient to dilute the Raymond effect spatially). This is a clear limitation of the steady-state assumption when modeling the age of the ice in the vicinity of a dome. A discussion on this effect is mandatory.*

Reply: Elmer-ice does in fact include all the physics that explains the Raymond effect. But a Raymond effect is not seen on the observed radar profiles.

The modeled Raymond effect is also quite obscure. Fig. R1 shows the modelled age against normalized depth on a transect perpendicular to the ridge (see the red straight line on the left plot). There are two rises in Fig. R1B, at about x=1015 and at x=1035. However, the depths at x<1015 should be much deeper according to Fig. 4 where the model overestimates ages at depths compared with the radar observation of 153.3 ka isochrone in triangle 3. The x=1035 feature might be a weak Raymond bump.

[Figure]

Fig R1. A) Bedrock elevation in the $70\times70$ km$^2$ region. The black lines are the route of the polarimetric radar in respect of the 153 ka isochrone plotted in Fig. 4; the common point of the four triangles is Dome A; the white cross is Kunlun station. B) the simulated age/normalized depth plot along the route marked in red in A). The isochrones contour interval is 100 ka.

We add a discussion on this effect in Uncertainties section as follows:

The special ice flow conditions at ice divides often leads to the presence of Raymond arches (Raymond, 1983), where older ice is at shallower depth than it is several ice thicknesses away from the divide. These features are visible as uplifted radar internal reflections in profiles across the divide. The strongest Raymond arches show up in high-accumulation coastal domes where the bed is cold and flat and the ice column is closer to isothermal (e.g. Hindmarsh et al., 2011). However, bed topography is complex at Dome A and Raymond arches are not seen in the observed radar profiles. Furthermore, our ice dynamics package, Elmer-ice, includes all the physics needed to produce the Raymond effect, but we also detect no such feature in transects across the flow divide. We explain this by the Raymond arch being obscured by a combination of rugged basal topography and thermal structure. The strong thermal gradient in the ice sheet tends to reduce the Raymond effect: the tendency of the non-Newtonian rheology to produce a stiff layer near the bed where strain rates are low is counteracted by the tendency of warm temperatures to produce softer ice at depth. The viscosity of the basal ice under the dome is softer than the viscosity of the super cold ice near the surface, but it is still much stiffer than the basal ice away from the dome, causing the old ice to be up-warped somewhat under the ridge. Moreover, the high basal melt rates of 2-3 mm a$^{-1}$ at Kunlun station draws down ice and obscure the Raymond effect.

*- Why is the age model compared to the radar age observations only at Kunlun? The comparison could be done anywhere where there are radar data.*

Reply: There are only dated isochrones along the 2 radar lines that connect Dome A with Vostok. We show the lines now in the new Fig. 1A. The rest of the AGAP, polarimetric and Chinare radar data we use in the paper is not tied to the Vostok ice core and hence no age-depth models exist along those radar lines. The most relevant use of the results is at the

location of the deep ice core site, which is the focus the simulation using the ice fabric taken from there.

[Figure]

Fig. 1 (A) The locations of Dome A (black circle), Kunlun (black +), Vostok (black ×) and the 70×70 km² study region (black box). The background is surface elevation with 100 m contour interval. The two radar lines that connect the Vostock drill site to Dome A are shown in red. (B) The 70×70 km² finite element mesh in the vicinity of Dome A projected on a polar stereographic map with standard parallel at 71°S and central meridian at 0°E. The background is bedrock elevation. The boundaries of the inner region and the whole region are shown, with the inner 30×30 km² region centred on Kunlun station has 300 m resolution, and the outer region 3 km resolutions. There are 21 terrain-following vertical layers with thinner layers near the base. The bar in the center denotes the drilling site at Kunlun station.

*- if I understood correctly, to compare the modeled vertical velocity with the accumulation observations, the authors use an average accumulation which is calculated as a weighted average of glacial and interglacial accumulations. This is too rough an hypothesis. The authors should use the EPICA Dome C record to calculate a ratio between the present-day accu and the 800 kyr average accu. This way, the comparison with the modeled vertical velocity would be more relevant- in a similar way, the authors should use a 800 kyr average value of the surface temperature based on the Dome C temperature variations (assuming the variations are the same at Dome A), rather than simply the present-day value at Dome A.*

Reply: Yes, this is true for accumulation and we modify the text thus:

The average accumulation during the past 800 ka is 17.7 mm i.e. a⁻¹ using the EPICA Dome C record (Bazin et al, 2013), which is very close to what the three best fit simulations achieve (Fig. 3B; Table 1).

However, we do not think it is better to use the 800 ka average value of the surface temperature based on the Dome C temperature variations than simply the present-day value at Dome A, because the important thing for the ice dynamics is how the viscosity of the ice would change over time and that is not a linear function of temperature.

In our simulations published in Sun et al. (2014), we tried both the present-day temperature (-58.5 °C) and that in glacial period (-68.5 °C).  The cold temperatures produce very poor fits that must be rejected. We explain in the text

The present-day surface temperature is −58.5°C, while it is likely about 10°C warmer than that during the Last Glacial Maximum (LGM) over the East Antarctic plateau (Ritz et al., 2001). The viscosity of the ice would change over time and that is not a linear function of temperature. Sun et al. (2014) found that none of the simulations using a surface temperature of -68.5°C matched well with the dated radar isochrones at Kunlun station, and we confirm that with the extended set of dated isochrones extending to 2/3 ice depth. While glacial period temperatures were likely warmer on average than -68.5°C they were certainly colder than present day. Sun et al. (2014) explain the poor fits for cold surface temperature simulations as being due to key role of warm interglacials in determining the vertical velocity profile of the ice because of the exponential Arrhenius dependence on temperature of the ice viscosity (Eqn (8)), along with much higher accumulation rates during interglacials. Therefore we prescribe surface temperature to be the present value of -58.5°C in this study.

*- there is a mistake at the beginning of section 4.1. At steady-state, surface vertical velocity should be equal to surface accumulation rate, not surface accumulation rate plus basal melting. This should be corrected.*

Reply: Yes. We agree with the referee. We made a mistake here. At steady-state, surface vertical velocity equals surface accumulation rate, while vertical velocity at the bottom equals the basal melting rate. We correct it in the revision.

At steady-state, surface vertical velocity equals surface accumulation rate. The average accumulation during the past 800 ka is 17.7 mm i.e.a$^{-1}$ using the EPICA Dome C record (Bazin et al, 2013), which is very close to what the three best fit simulations achieve (Fig. 3B; Table 1).

*- Because of these rough assumptions in the modeling, a perspective paragraph listing what could be improved in a future modeling study would be welcome.*

*Clear suggestions for improving the model include: Non-steady state dynamics; Basal hydrology allowing for water flow and refreezing*

Reply: In the revision, we add a substantial section of Uncertainties listing what could be improved in a future modeling study.

Our approach here is relatively sophisticated in terms of ice models presently in use, but there are several limitations that almost certainly mean that details of the simulation will be wrong. We make the key assumption that the ice sheet is in steady-state, and the surface geometry is fixed, which means the surface accumulate rates balances the vertical velocity and it is also fixed in time. However, the basal thermal condition is sensitive to the ice thickness although other simulations of the whole Antarctic ice sheet suggest that elevation changes at Dome A have been less than 50 m over glacial cycles (Ritz et al., 2001; Saito and Abe-Ouchi, 2010.) Transient simulations with varying geometry and surface accumulation rate in the past 800 ka would improve the model result.

We used a spatially constant geothermal heat flux. Although geothermal flux may over kilometer scales, it seems unlikely in East Antarctica. For example, Carson et al., (2014)

suggest heat flow may vary by a factor of >150% over 10–100 km length scales in East Antarctica. Passalacqua et al., (2017) explored variation in heat flux around Dome C using data from radar surveys, and prescribe uniform geothermal heat flux over 10 km scales. Schroeder et al (2014) similarly infer geothermal heat flux variability from radar surveys over Thwaites glacier in West Antarctica, which is proximal to the Mount Takahe volcano that was active during the Quaternary, finding heat fluxes could double over ranges of about 20 km. We do not expect any recent magmatic activity in the Gamburtsev Mountains, and the situation of Dome C is probably a reasonable analogue. However there is simply no data to constrain heat flux around Dome A, and hence modelled thermal structure, ice viscosity and age-depth profile. Liefferinge and Pattyn (2013) explored the uncertainty in existing geothermal heat flux data sets and their effect on basal temperature with a spatial resolution of 5 km. The basal temperature was calculated using the steady-state thermodynamic equation in which ice flow velocity is calculated from the shallow-ice approximation. The mean geothermal heat flux of the three existing datasets at Dome A is about 45 mWm$^{-2}$, with root mean square error of about 20 mWm$^{-2}$. Their modelled basal temperature at Dome A is about -10°C corrected for the dependence on pressure with a root mean square error of about 6°C. Due to the coarse resolution (5 km) used in the whole Antarctic simulations of Liefferinge and Pattyn (2013), the modelled basal temperature does not have obvious spatial variation across the Dome A region at scales of hundreds kilometers.

The Gamburtsev Mountain is characterized by large spatial variability in bedrock topography, which means that a full-Stokes model that considers the all the stress components is better able to capture the ice dynamics than does the shallow-ice approximation (e.g., Zhao et al., 2013). In our study, large variations in basal temperature are simulated using a full-stokes model run at around 500 m resolution. The basal thermal state is then very sensitive to geothermal heat flux (Sun et al., 2014), which we explored using 45, 50, 55 and 60 mWm$^{-2}$, and which spans the broad range suggested by Liefferinge and Pattyn (2013).

We also use a spatially constant fabric across all our model domain, with transitions between fabrics at two fixed depths taken from those measured at Kunlun station by Wang et al., (2017). As discussed in Section 4.2, this leads to lower confidence in the age of the basal ice in the region south of Kunlun than to the north. This further means that we have more confidence in finding very old ice in the slightly further away northern region of Fig. 6 than to the south of Kunlun.

Our results suggest spatial variability in basal melting, and this may introduce basal accretion in places (Bell et al., 2011), though there is no radar evidence of any basal accretion features in the vicinity, the model could be improved by adding basal hydrology. Basal melting may also introduce sliding at the ice/bed interface, which we explicitly excluded in the model, however, comparison with observed horizontal velocities suggests that this is not an issue. Indeed extraction of sliding rates from inverse modeling using observed velocities would be extremely difficult at Dome A given the very low speeds making satellite interferometry impossible, and the sparse network of GPS locations.

*I also have some minor points below:*

*- l.51: "Hou et al., 2007" -> missing space*

Reply: Done.

*- l.102: "special" -> "spatial"*

Reply: Done.

*- Fig.1A is difficult to ready. I would use a square region in a classical projection.*

Reply: OK. The other referee also said Fig.1 A is not informative. So we change it to show important feature such as the radar profiles connecting Kunlun and Vostok.

*- l.165: "is gas constant" -> "is the gas constant"*

Reply: Done.

*- l. 173: "the components of..." -> missing space*

Reply: Done.

*- l. 242: why not using an intermediate value of the surface temperature between the present-day and the LGM? (Cf. comment above).*

Reply: Sun et al. (2014) used surface temperatures both in present-day value (-58.5 °C) and that in glacial period (-68.5 °C). It is clear that -68.5 °C (the full glacial temperatures) cannot produce a good match to the internal reflection horizons and vertical velocities.

*- l245: the no sliding assumption is quite rough. There is probably sliding where there is melting.*

Reply: It is possible there is sliding. Although surface speeds in our study region is small (a mean speed of ~11±2.5 cm a$^{-1}$), and well matched to the model results we find from ice deformation without basal sliding (Fig. 7), hence basal sliding must be a small fraction of the total velocity, and not affect the results we show. Attempting an inversion from observation velocities would introduce very large errors because the speeds are below the error margin from satellite measurements and thus only available from the very sparse GPS network shown in Fig. 7.  We mention this both in Section 3.4:

We run the model with a no-slip condition at the bed. We could expect that sliding might occur where there is melting at the bottom. However, surface speeds in our study region is very small (a mean speed of ~11±2.5 cm a$^{-1}$, Yang et al., 2014) and well matched to the model results we show later from ice deformation without basal sliding (Section 4.4) hence basal sliding must be a small fraction of the total velocity, and not affect the results we show.

 And in the discussion:

Our results suggest spatial variability in basal melting, and this may introduce basal accretion in places (Bell et al., 2011), though there is no radar evidence of any basal accretion features in the vicinity, the model could be improved by adding basal hydrology. Basal melting may also introduce sliding at the ice/bed interface, which we explicitly excluded in the model, however, comparison with observed horizontal velocities suggests that this is not an issue. Indeed extraction of sliding rates from inverse modeling using observed velocities would be extremely difficult at Dome A given the very low speeds making satellite interferometry impossible, and the sparse network of GPS locations.

*- l. 261: quite a big assumption here also, since the geothermal flux might change at a kilometer scale.*

Reply: Actually we note that other studies in Antarctica expect no large variations over 10 km scales. But in principle this is a problem, and we discussion in the Uncertainties:

We used a spatially constant geothermal heat flux. Although geothermal flux may over kilometer scales, it seems unlikely in East Antarctica. For example, Carson et al. (2014) suggest heat flow may vary by a factor of >150% over 10–100 km length scales in East Antarctica. Passalacqua et al. (2017) explored variation in heat flux around Dome C using data from radar surveys, and prescribe uniform geothermal heat flux over 10 km scales. Schroeder et al (2014) similarly infer geothermal heat flux variability from radar surveys over Thwaites glacier in West Antarctica, which is proximal to the Mount Takahe volcano that was active during the Quaternary, finding heat fluxes could double over ranges of about 20 km. We do not expect any recent magmatic activity in the Gamburtsev Mountains, and the situation of Dome C is probably a reasonable analogue. However there is simply no data to constrain heat flux around Dome A, and hence modelled thermal structure, ice viscosity and age-depth profile. Liefferinge and Pattyn (2013) explored the uncertainty in existing geothermal heat flux data sets and their effect on basal temperature with a spatial resolution of 5 km. The basal temperature was calculated using the steady-state thermodynamic equation in which ice flow velocity is calculated from the shallow-ice approximation. The mean geothermal heat flux of the three existing datasets at Dome A is about 45 mWm$^{-2}$, with root mean square error of about 20 mWm$^{-2}$. The modelled basal temperature at Dome A is about -10°C corrected for the dependence on pressure with a root mean square error of about 6°C. Due to the coarse resolution (5 km) used in the whole Antarctic simulations of Liefferinge and Pattyn (2013), the modelled basal temperature does not have obvious spatial variation across the Dome A region at scales of hundreds kilometers.

The Gamburtsev Mountain is characterized by large spatial variability in bedrock topography, which means that a full-Stokes model that considers the all the stress components is better able to capture the ice dynamics than does the shallow-ice approximation (e.g., Zhao et al., 2013). In our study, large variations in basal temperature are simulated using a full-stokes model run at around 500 m resolution. The basal thermal state is then very sensitive to geothermal heat flux (Sun et al., 2014), which we explored using 45, 50, 55 and 60 mWm$^{-2}$, and which spans the broad range suggested by Liefferinge and Pattyn (2013).

*- l. 326: the reference is Ruth et al., not Urs et al. (Urs is the first name).*

Reply: Done. Thanks for that info!

References:

Bazin, L., Landais, A., Lemieux-Dudon, B., Kele, H. T. M., Veres, D., Parrenin, F., Martinerie, P., Ritz, C., Capron, E., Lipenkov, V., Loutre, M.-F., Raynaud, D., Vinther, B., Svensson, A., Rasmussen, S. O., Severi, M., Blunier, T., Leuenberger, M., Fischer,

H., Masson-Delmotte, V., Chappellaz, J., and Wolff, E., 2013. An optimized multi-proxy, multi-site Antarctic ice and gas orbital chronology (AICC2012): 120–800 ka. *Climate of the Past*, **9**, 1715–1731.

Bell, R. E., Ferraccioli, F., Creyts, T. T., Braaten, D., Corr, H., Das, I., Damaske, D., Frearson, N., Jordan, T., Rose, K., Studinger, M., and Wolovick, M.: Widespread Persistent Thickening of the East Antarctic Ice Sheet by Freezing from the Base, Science, 331, 1592–1595, doi:10.1126/science.1200109, 2011.

Carson, C. J., McLaren, S., Roberts, J. L., Boger, S. D., Blankenship, D. D.: Hot rocks in a cold place: high subglacial heat flow in East Antarctica, J. Geol. Soc., 171, 9–12, https://doi.org/10.1144/jgs2013-030, 2014.

Ritz, C., Rommelaere, V., and Dumas, C.: Modeling the evolution of Antarctic ice sheet over the last 420,000 years: implications for altitude changes in the Vostok region, J. Geophys. Res., 106, 31943–31964, 2001.

Passalacqua, O., Ritz, C., Parrenin, F., Urbini, S., Frezzotti, M.: Geothermal flux and basal melt rate in the Dome C region inferred from radar reflectivity and heat modelling, The Cryosphere, 11, 2231-2246, https://doi.org/10.5194/tc-11-2231-2017, 2017.

Saito, F. and Abe-Ouchi, A.: Modelled response of the volume and thickness of the Antarctic ice sheet to the advance of thegrounded area, Ann. Glaciol., 51, 41–48, 2010.

---

## Author Comment (AC2) · 3 Apr 2018

In the reply, the referee's comments are in italics, our response is in normal text, and quotes from the manuscript are in blue.

*Review of Where is the 1-million-year-old ice at Dome A? Liyun Zhao, John C. Moore, Bo Sun, Xueyuan Tang and Xiaoran Guo*

*The study aims to locate areas around Kunlun Station that contain ice older than 1Myr, and the manuscript describes results from simulations with the Elmer/Ice model, a 3D, thermomechanical, Full-Stokes, finite-element model. The model parameters are constrained by observations of surface velocity, ice fabric and radar isochrones. The authors test different ice fabrics and geothermal heat fluxes and conclude through the comparison with observations that the geothermal heat flux is likely between $55mW/m^2$ and $60mW/m^2$. From the model results, areas that potentially contain ice older than 1Myr are identified based on these two geothermal heat flux values. Overall, the manuscript is well written and is definitely of interest to the glaciological community especially those concerned with ice-core sites and the retrieval of "the oldest ice". I have some main points that I would like to see addressed, as well as some minor points that might improve readability.*

***Main points***

*1 I am missing a more thorough discussion of the uncertainties or rather the confidence the authors have in their results (Figs. 6 and 8). Firstly, a comparison with the results from van Liefferinge and Pattyn (2013) would be interesting in order to give insights into how coarse vs. fine grid, and shallow-ice vs. full-Stokes might affect results.*

Reply: We add a discussion of the uncertainties in an extended **uncertainties** section, including comparison with Liefferinge and Pattyn (2013), and other assumptions and limitations:

[revised manuscript text omitted]

*(2) Secondly, the vertical distribution of ice fabric is not in line with the findings of Wang et al. The authors state that they obtain the same results for two or four layers of fabric but the same what? Age-depth distribution? Surface velocity? 153.3kyr isochrone depth?*

Reply: Actually the data in Wang et al. (2017) are problematic in some parts of the survey they do detect coherent signals from 6 or even 4 layers. Experiments with 4 layers T1:T4 show a slightly larger vertical velocity at Kunlun station and deeper depth of 153.3ka isochrones, and almost the same horizontal flow velocity than as with just the top two layers T1 and T2 in our simulations.

We note this in Section 3.3:

The Wang et al. (2017) lowermost T5 and T6 layers are rather weak and indistinct in most of the survey grid. The layer T1 is relatively flat, T2 are relatively flat in half region and has large spatial variation of depth in the other half, while the T3 and T4 layers have large spatial variation of depth and are even missing in some locations around their survey grid. Experiments with 4 layers T1:T4 show a slightly larger vertical velocity at Kunlun station and deeper depths for the 153.3ka isochrone (Section 4.2), and almost the same horizontal flow velocity as with just the top two layers T1 and T2 in our simulations. So here we present simulation results based on a fabric model using just the two upper layers.

*Based on the figures in Wang et al. (2017), I would say that an assumption of two layers is fine around Kunlun station but does not appear to be valid further away from the station. Could the variability of T3 and T4 explain the variation in fit with the isochrone? Finally, the fact that the model has a tendency to underestimate the depth of the 153.3kyr isochrones indicates that the model could be overestimating the amount of 1Myr old ice (with the possible exception of triangle 4). Especially triangle 3 is problematic and perhaps by extension the age of the valley.*

Reply: Yes we agree this needs emphasizing in the paper. In fact, the variability of T2, T3 and T4 explains all.

Wang et al. (2017) shows that the depth of T2 layer is relatively constant in most of the region to the north of Kunlun Station, but is much deeper to the south. As does the depth of T3.

A deeper depth for T2 would result in larger vertical ice velocity, hence a deeper depth of the 153.3ka isochrones. Triangle 4 is well inside the north part, moreover, it is closest in depths of the T2-T4 layers to those at Kunlun station (see Fig. 9 in Wang et al., 2017). Therefore, the modeled depths of the 153.3ka isochrones in triangle 4 is as most confidence among the four triangles (Fig. 4). In contrast, triangles 2 and 3 are in the southern region, therefore, the modeled depths of the 153.3ka isochrones are underestimated (Fig. 4). This magnitude of this underestimation depends on the ice

thickness. The underestimation is larger where the ice is thicker (triangle 3) and smaller in triangle 2 where the ice is thinner. This explains why triangle 3 is problematic.

In the text, we add this in section 4.2:

Wang et al. (2017) observed large spatial variability of the depth of the ice fabric layers T2:T4. As noted in Section 3.3, we applied the depth of the top two fabric layers T1:T2 at Kunlun station to the whole region. Wang et al. (2017) show that the depth of the T2 layer is relatively constant in the region to the north of Kunlun Station (which includes triangle 4 in Fig. 4A), but is much deeper in the southern region (including triangles 2, and 3). Smaller T2 layer depth would result in slower simulated ice vertical velocity, hence older ice. Therefore, the modeled depths of the 153.3ka isochrones to the south of Kunlun are underestimated. The underestimation is larger where the ice is thicker (triangle 3) and smaller in triangle 2 where the ice is thinner, see Fig. 4B.

We also add this sentence in Section 4.3

As we discussed earlier in section 4.2, the age in triangle 3 are probably overestimated (Fig. 5C and D). The modeled age inside triangle 4 has the most confidence.
and
There is ice at least 1 million years old ice simulated on the side slopes of the valley below Kunlun station. The closest to Kunlun station being found directly below a point about 380 m away under 55 mW m$^{-2}$, and 1 km away under 60 mW m$^{-2}$ heat fluxes. However this position is a less reliable part of the domain than the area to the north of Kunlun in triangle 4, where old ice is about 5 km away.

We add the four triangles in Fig.s 5 and 6. The modeled age inside triangle 4 is the most reliable.
We also note this in the Uncertainties
We also use a spatially constant fabric across all our model domain, with transitions between fabrics at two fixed depths taken from those measured at Kunlun station by Wang et al. (2017). As discussed in Section 4.2, this leads to lower confidence in the age of the basal ice in the region south of Kunlun than to the north. This further means that we have more confidence in finding very old ice in the slightly further away northern region of Fig. 6 than to the south of Kunlun.
And Conclusion
Using our favored values for geothermal heat flux and ice fabric, ice this ancient may be found by vertical drilling within 400 m of the present Kunlun drill site, indeed this location would contain much older ice since it seems to be frozen to the bed. However we have more confidence in our simulation of ancient ice about 5-6 km to the north of Kunlun station than the closer sites to the south. Near-basal ice this close to Kunlun may be accessible with a straight forward repositioning of the drilling site (Talalay et al., 2017) rather than the logistics base.

*2 Most of the figures are small and hard to interpret. Please add legend and/or title so*

*that main points of the figures are understandable without consulting the figure caption. The symbol for Kunlun Station, the letters distinguishing the figures and the axes labels are hard to read.*

Reply: Actually we followed the submission guidelines of The Cryosphere, but we try to improve the figures as much as we can. We change the size of some figures to make it larger to see in the revision. The figures are small also due to the format of TCD – it is not 100% occupied. We expect the figures will be larger and higher resolution in the final TC version. We add letters of "Kunlun" in Fig. 4b and Fig. 5-8, and distinguishing legends in Fig. 5-7. We make a new Fig. 1A.

*3 The structure of the paper could be improved. For example, the introduction and the discussion/conclusion appear fragmented rather than cohesive sections (e.g., the introduction contains a section about Kunlun, then a section about ice fabric and vertical velocities and then another section about Kunlun). I am also missing an introduction to the datasets in Section 2. The surface velocity data, the polarimetric data and the isochrones depths should be mentioned in Section 2 before they are used in Section 4.*

Reply: Yes, we agree. The introduction has been reorganized. We have also split and expanded the Uncertainties and Conclusions. We have expanded the Section 2 (Domain, Data and Mesh):

The surface and topography ice thickness in the Dome A region come from both airborne and ground based measurements. The Antarctic Gamburtsev Province Project (AGAP) surveyed the region with flight lines 5 km apart orientated in the north-south direction (Bell et al., 2011) with perpendicular lines every 33 km. Ground-based surveys were done by the 21st and 24th Chinese National Antarctic Research Expedition (CHINARE) in a 30×30 km square along lines typically a few km apart, the along-track radar resolution is 125 m (Sun et al., 2009; Cui et al., 2010). A stake network was also established for ice motion using differential GPS receivers, and data collected in 2008 and 2013 (Yang et al. (2014). More recently polarimetric radar observations were also collected on 5 km-spaced ground-based survey grid (Wang et al., 2017) using a 179 MHz radar using orthogonal orientated antennae with 17 m along-track spacing. They deduced the existence of 4-6 layers of different ice fabric in their survey, and we make use of these data for the ice dynamics simulation.

The Vostok ice core provides absolute dates for radar internal reflections, and since these radar reflections are often continuous over hundreds of kilometers that can provide age-depth profiles over an extensive region of the ice sheet (e.g. Wang et al., 2017). Isochrones in two 150 MHz airborne radar transects collected by the Alfred Wegener Institute were tracked from Vostok to Dome A (Sun et al., 2014; Wang et al., 2017; Fig. 1A) providing 12 dated layers at Kunlun station. We select the 153.3ka isochrones for detailed analysis making use of polarimetric radar data collected in a set of 4 triangles centred on Dome A and 160 km in length.

***Minor points***

*Lines 13-33: The abstract should also state the aim of the study.*

Reply: We add to the abstract:

To investigate the age-depth profile to be expected of the on-going deep ice coring at Kunlun station, Dome A

*Line 13 (and other places): age/depth or age-depth?*

Reply: we use age-depth uniformly in the revision.

*Line 25 and 313: Typo "polarimetric"*

Reply: We corrected them.

*Lines 35-42: This part of the introduction needs a few more sentences about the background for the search a 1Myr ice, for example, with references to the IPICS white paper and to Fisher et al., 2013 (https://www.clim-past.net/9/2489/2013/)*

Reply: Thank you. We change this part of introduction as

Finding a continuous and undisturbed million-year old ice core record in Antarctic has been identified by the International Partnership for Ice Core Sciences (IPICS) as one of the most important scientific challenges in ice core research in the near future (http://www.pages.unibe.ch/ipics/white-papers). This is because the last 8 glacial cycles are characterized by irregular cycles of roughly 100 ka in length. Both climate and greenhouse gases co-vary closely. Between 900 ka and 1.2 Ma BP glacial cycles are more regular and are paced at significantly higher frequencies (Lisiecki and Raymo, 2005). The relationship between greenhouse gases and ice sheet growth and decay during these times is presently unknown since it can only be derived from the atmospheric record archived in an Antarctic ice core covering this time interval.

The search for a continuous and undisturbed stratigraphic record containing 1 million-year old ice has also interested and challenged the ice modeling communities for several decades (e.g., Van Liefferinge and Pattyn, 2013).

In the model, the inner $30 \times 30$ km$^2$ region centred on Kunlun station has 300 m resolution, and the outer region 3 km resolutions. So the resolution of the model is comparable with the radar data in the central part of the domain that we focus on.

*Line 130: Word missing?*

Reply: We add the word "that". Now it reads

where τ is the deviatoric stress tensor that has a non-linear constitutive relationship with the strain rate tensor

*Line 131: Add that the strain rate tensor will be discussed in more detail below.*

Reply: Done.

*Line 145: Typo: "Gudmundsson"*

Reply: Done.

*Line 157: what is n equal to?*

Reply: n equal to 3. We add it in the revision.

*Lines 160-165: These values might be better summarized in a table*

Reply: Actually we tested this, but we think that it is more convenient for the readers if the values are shown in the text.

*Section 4: I would like a table in this section where different simulations are listed with least-square age error, vertical surface velocity, age at the bedrock and the magnitude of the difference between measured and modelled horizontal velocity. This would allow readers to judge the difference between the model results and how well they agree with observations.*

Reply: OK. We add this table (Table 1) in the revision.

Table 1. Modeled and observed isochrones ages and surface velocities at Kunlun.

| Simulation | Isochrone age RMS error (ka) | Modeled-observed vertical surface velocity (mm a$^{-1}$) | Horizontal speed RMS error (cm a$^{-1}$) | Modeled age at bedrock (ka) |
|---|---|---|---|---|
| Kunlun fabric, G=55 mW m$^{-2}$ | 10.85 | -0.1 | 4.9 | 831 |
| Kunlun fabric, G=60 mW m$^{-2}$ | 26.00 | 4.2 | 2.7 | 649 |
| Girdle fabric $z_s = 1/3$, G=60 mW m$^{-2}$ | 6.88 | -1.2 | 3.9 | 687 |
| Girdle fabric $z_s = 2/3$, G=50 mW m$^{-2}$ | 27.18 | -0.9 | 6.9 | 1143 |

*Lines 360-364: explicitly refer to the figure that is being discussed (5C, 5D etc.) otherwise it is hard to follow.*

Reply: Done.

*Line 402: I agree with reviewer #1 re. the comparison with accumulation + basal melt rate.*

Reply: We agree with the referee. We made a mistake here. At steady-state, surface vertical velocity equals surface accumulation rate, while vertical velocity at the bottom equals the basal melting rate. We correct it in the revision.

The modeled surface vertical velocity distribution is shown in Fig. 7B and, as discussed earlier (section 4.1), may be compared with local accumulation.

*Line 442: please indicate on a figure where this valley is.*

Reply: We add the outline of the valley (the contour of 1500 m in bedrock elevation) in Fig. 8A&B – the modeled basal temperature plot.

*Line 455: The hydraulic potential should be shown on a figure or in the supplementary material. The description here does not help the reader.*

Reply: OK. We add this plot in the Fig. 8C in the revision.

[Figure]

Fig. 8 Basal temperature relative to pressure melting point using Kunlun fabric and a geothermal heat flux of 55 (A) and 60 (B) mW m$^{-2}$, and the hydraulic potential (C). The bedrock areas at pressure-melting point in A and B are surrounded by a white contour. Kunlun station is marked as white plus sign. The black curve in A and B shows the outline of the valley defined as the 1500 m contour in bedrock elevation.

*Line 522: Considering that Dome A has very low accumulation rates, how can an ice core from the area provide the highest resolution record?*

Reply: We are comparing to other stratigraphic records such as sediment cores. However, we delete "and highest resolution" in this sentence

Thus the Kunlun station is well suited to provide the longest continuous stratigraphic record from Antarctica.

*Fig. 1: Both figures appear pixelated and especially 1A is not very informative.*

Reply: The other referee also complained about Fig. 1. We revised Fig. 1A to show more useful information such as the radar lines connecting to Vostok. The pixilation issue is probably something to do with how TC makes the pdf. In fact when we tested it we see no issues zooming to 500%, but anyway it should be improved in the publication version.

*Fig. 2: this figure is hard to read, why not split it into two figures?*
Reply: OK. We split it into two plots.